# Stable Convergent Polyneuronal Innervation and Altered Synapse Elimination in *Orbicularis oculi* Muscles from Patients with Blepharospasm Responding Poorly to Recurrent Botulinum Type-A Neurotoxin Injections

**DOI:** 10.3390/toxins16120506

**Published:** 2024-11-24

**Authors:** Brigitte Girard, Aurélie Couesnon, Emmanuelle Girard, Jordi Molgó

**Affiliations:** 1Service d’Ophtalmologie, Hôpital Tenon, Sorbonne Université, Université Pierre et Marie Curie, 4 rue de la Chine, 75020 Paris, France; docteur.girard@orange.fr; 2Hôpital Privé Armand Brillard, 3. Avenue Watteau, 94130 Nogent sur Marne, France; 3Institut des Neurosciences Paris-Saclay, UMR 9197, CNRS/Université Paris-Sud, 91198 Gif-sur-Yvette, Cedex, France; 4Institut NeuroMyoGene—Physiopathology & Genetic of Neuron and Muscle, CNRS UMR5261, INSERM U1315, Université Lyon1, 8 Avenue Rockefeller, 69008 Lyon, France; emmanuelle.girard@univ-lyon1.fr; 5Plateforme Aniphy, SFR Santé Lyon-Est, CNRS UAR3453, INSERM US7, Université Lyon1, 8 Avenue Rockefeller, 69008 Lyon, France; 6Service d’Ingénierie Moléculaire pour la Santé (SIMoS), EMR CNRS 9004, Département Médicaments et Technologies pour la Santé (DMTS), Institut des Sciences du Vivant Frédéric Joliot, Commissariat à l’énergie Atomique et aux Énergies Alternatives (CEA), Université Paris-Saclay, 91191 Gif-sur-Yvette, France

**Keywords:** blepharospasm, botulinum type-A neurotoxin, convergent innervation, human *Orbicularis oculi* muscle, myectomy, neuromuscular junction, nerve sprouting, nicotinic acetylcholine receptors, polyneuronal innervation, synapse elimination, skeletal muscle

## Abstract

Botulinum neurotoxin type-A (BoNT/A), which blocks quantal acetylcholine (ACh) release at the neuromuscular junction (NMJ), has demonstrated its efficacy in the symptomatic treatment of blepharospasm. In 3.89% of patients treated for blepharospasm at Tenon Hospital, BoNT/A was no longer effective in relieving the patient’s symptoms, and a partial upper myectomy of the *Orbicularis oculi* muscle was performed. We used surgical waste samples from 14 patients treated with repeated injections of either abobotulinumtoxinA (Dysport^®^) or incobotulinumtoxinA (Xeomin^®^). These muscle fragments were compared to others from 4 normal subjects, naïve of BoNT/A. The morphological study was performed blinded to the BoNT/A treatment and between treated and control samples. Neuromuscular specimens analyzed by confocal laser scanning microscopy, using fluorescent staining and immune-labeling of presynaptic proteins, revealed that the pattern of innervation (e.g., polyneuronal and convergent innervation), the muscle nicotinic ACh receptors (nAChRs), and the NMJs exhibited marked differences in BoNT/A-treated muscles (regardless of the toxin clinically used), with respect to controls. BoNT/A-treated junctions exhibited profuse polyneuronal innervation in which 2–6 axons innervated 74.84% of single muscle fibers, while 99.47% of control junctions were mono-innervated. Another new finding was the stable convergent innervation, in which several motor axons end onto the same endplate. Morphological signs of synapse elimination included the presence of retraction bulbs in axons and nerve terminals and a reduced extension of postsynaptic nAChRs. These outcomes suggest that synapse elimination is altered and raise questions on the origin and factors contributing to the plasticity changes observed and the functioning of NMJs.

## 1. Introduction

Botulinal neurotoxins, produced either by toxigenic anaerobic *Clostridium botulinum* strains and other *Clostridium* bacteria or by non-*Clostridium* species (reviewed in refs. [1,2,3,4,5]), are well known for their high toxicity and as a food safety hazard. However, despite their potent toxicity in humans and animals (reviewed in [6]), some of the serotoxin types, like botulinum neurotoxin type-A (BoNT/A), have become effective drugs for a remarkable number of therapeutic usages in different pathological indications and disorders, as well as in cosmetics (reviewed in [7,8,9,10,11]). Currently, a number of movement disorders are treated with BoNT/A (reviewed in refs. [12,13,14,15,16,17]). BoNT/A is known to target presynaptic cholinergic motor nerve terminals of the skeletal neuromuscular junction (NMJ) and to block spontaneous and nerve-evoked quantal acetylcholine release leading in vivo to a flaccid skeletal muscle paralysis (reviewed in [18,19,20]). Despite the long-term transient paralysis of skeletal muscle caused by BoNT/A (from a few weeks duration in rodents to several months in humans), the original nerve terminals and perisynaptic Schwann cells are maintained at vertebrate NMJs [21,22].

Significant progress has been made in understanding the molecular mode of action of BoNT/A. The modular polypeptide structure of BoNT/A consists of a light chain (L, 50 kDa) and a heavy chain (H, 100 kDa) linked by a single disulfide bond and folded in four molecular domains. Each of these domains plays a specific role in the five main steps involved in the mode of action of the toxin, as reported in a number of outstanding reviews that the reader is invited to consult [2,8,9,23,24,25]. For the purpose of this article, we will simply recall that the steps involved in BoNT/A action include: (i) Binding to specific membrane receptors in the nerve terminal membrane (polysialogangliosides and the synaptic-vesicle glycoprotein protein 2 (SV2)) [26,27,28]. (ii) Endocytosis of the toxin complex into the synaptic vesicle lumen. BoNT/A for synaptic vesicle entry requires a tripartite polysialoganglioside-synaptotagmin-1-SV2 plasma membrane nanocluster, which controls the endocytic sorting into synaptic vesicles [29,30]. (iii) Translocation of the L toxin domain to the nerve terminal cytosol. (iv) Reduction of the interchain disulfide bond and cytosolic release of the metalloprotease L chain. (v) Hydrolysis of a single peptide bond in its unique substrate in the C-terminal region of the synaptosomal associated protein of 25 kDa (SNAP-25) within the nerve terminal [31].

Among the distinct movement disorders, it is included the essential blepharospasm, a focal cranial dystonia characterized by bilateral involuntary sporadic or permanent eyelid closures caused by contractions of the *Orbicularis oculi* muscles. The severity of movements can range from an increased blinking frequency to significant functional blindness (for a recent review on the neural control of blinking, see ref. [32]. At present, the origin of this neurological disorder is not well understood, and no etiological treatment is available (reviewed in [33,34,35]). However, treatment with BoNT/A injection can effectively alleviate the symptoms of essential blepharospasm and has become the treatment of choice in the management of this disease condition, constituting the most common symptomatic medical treatment. The impressive response rate and the acceptable safety profile of BoNT/A injections for blepharospasm have been proven through multiple clinical observation studies and over long periods of time [35,36,37,38,39]. Due to the slow functional recovery of neuromuscular transmission after BoNT/A treatment in patients with blepharospasm, between six weeks and four months [40,41], the periodic BoNT/A administration every two to three months is compulsory to maintain a satisfactory spasm-free treatment. Regardless of this limitation, the toxin securely relieves patients of their dystonia symptoms and considerably enriches their quality of life [42,43]. However, in some patients with severe essential blepharospasm having persistent symptoms and failing to show a therapeutic response to BoNT/A-treatment, the upper eyelid surgical myectomy of a part of the upper *Orbicularis oculi* muscle is an effective recommended treatment of this pathological condition (reviewed in [44,45,46,47]).

The aim of the present study was to characterize morphologically the innervation profile of *Orbicularis oculi* muscles removed from patients with essential blepharospasm that have been treated with repetitive injections of BoNT/A and in which BoNT/A-treatment was no longer effective in controlling the recurrent spams and in relieving the patient’s symptoms. For this, we used surgical waste muscle specimens from 14 patients treated with repeated injections of either abobotulinumtoxinA (Dysport^®^) or incobotulinumtoxinA (Xeomin^®^). These muscle-fragment specimens were compared to others obtained during blepharoplasty in normal subjects that were naïve to BoNT/A. The morphological analysis, using confocal laser scanning microscopy, was blinded to the BoNT/A sample treatment used. Preliminary accounts of parts of this work have been presented in meetings and published in abstracts [48,49,50].

## 2. Results

### 2.1. Collection of Surgical-Waste Orbicularis oculi Muscle Fragments

At the Department of Ophthalmology of the Tenon Hospital in Paris, patients are being treated for benign essential blepharospasm with two commercially available preparations containing BoNT/A, either abobotulinumtoxinA (Dysport^®^) or incobotulinumtoxinA (Xeomin^®^). Interestingly, in fourteen patients of this population (3.89%), BoNT/A was no longer effective in reducing muscle spasms. It should be emphasized that in those patients, the loss of BoNT/A efficacy was not associated with the presence of neutralizing antibodies to the neurotoxin, as briefly reported previously [49,50].

The mean age of the 14 patients with blepharospasm to whom BoNT/A was ineffective was 69.4 years (range 53–76 years), females represented 64.3% and males 35.7%. The mean number of toxin injections delivered to patients before palpebral surgery was 17.4 (95% confidence limits 4.5–40) per treated subject. The points of BoNT/A injection are shown in Figure A1 (Section A.1). The mean time elapsed between repeated toxin injections was 10.6 weeks (95% confidence limits 8–12) per treated patient). The mean number of BoNT/A units used per injection was 350.1 ± 64.5 Speywood units for abobotulinumtoxinA (Dysport^®^) and 75.7 ± 9.96 Xeomin units for incobotulinumtoxinA (Xeomin^®^). All data was managed anonymously without knowledge of the surgical-waste muscle source specimen.

An overview of the surgical procedure used for partial upper *Orbicularis oculi* myectomy is given in Section A.1 and Section A.2 (Figure A1 and Figure A2). The aim of the partial upper myectomy was (i) to weaken the intensity of the blepharospasm, (ii) to increase the intervals between the BoNT/A injections, and (iii) to reduce the dosage of neurotoxin used.

### 2.2. Pattern of Innervation of Orbicularis oculi Muscle in BoNT/A-Treated Patients

The immuno-fluorescence labeling of neurofilaments allowed the visualization of axons and nerve terminals in *Orbicularis oculi* mount-muscle fragments (Figure 1A). The concomitant staining with α-bungarotoxin conjugated with fluorescein labeled the nAChRs, revealing the muscle endplates, as shown in Figure 1B. However, at low magnification, it was difficult to determine the innervation pattern at the single neuromuscular junction (NMJ) level.

For that reason, the neuromuscular specimen shown in Figure 1 was further imaged and analyzed at higher magnification, revealing the individual endplates and their individual motor axons and nerve terminals. As shown in Figure 2, a conspicuous characteristic of BoNT/A-treated muscles was the presence of a profuse and stable polyneuronal innervation that was typified by an unambiguous feature, the presence of distinct motor axons converging to a unique endplate (convergent innervation), as shown in Figure 2 (Ca, Cb, Cc).

Other examples of stable polyneuronal innervated NMJs are depicted in Figure 3 and Figure 4. The ability of axons and nerve terminals to sprout new neuronal processes is not lost during the repeated injections of BoNT/A, as shown in Figure 3B,b. In fact, sprouting was abundant in BoNT/A-treated muscle specimens from patients having received the last toxin injection a few days to four months before the myectomy of the *Orbicularis oculi.* The fluorescent staining and immuno-labeling of presynaptic proteins revealed the presence of small and long sprouts. In general, these outgrowths come into view as thin filament sprouts of variable length (2.2–40 µm length), emerging from axonal regions (nodal sprouts), sometimes reaching neighboring muscle fibers (collateral sprouts). Other sprouts originate from nerve terminals (ultra-terminal sprouts), and most remain within the same muscle fiber, as shown in Figure 3B,b. Axonal outgrowth was profuse in muscles from patients injected during several years with BoNT/A to the point that it was difficult to follow the motor innervation of single neuromuscular junctions.

In another frequent observation of polyinnervation, multiple axons innervated different sites of single muscle fibers so that the same muscle fiber had different endplates separated by tens of micrometers. A typical example is shown in Figure 5.

### 2.3. Pattern of Innervation of Orbicularis oculi Muscle from Control Patients

The control NMJs in *Orbicularis oculi* muscle fragments were always innervated in a clear-cut manner by the motor axon terminal, which is usually located halfway along the length of the fibers. Along these lines, NMJs were relatively simple and constantly mono-innervated by a single motor axon. However, great variability in the junction’s extension and shape outline was observed, as revealed by the typical nAChR array labeled with fluorescent α-bungarotoxin.

Examples of control human NMJs are shown in Figure 6A–F. In contrast to BoNT/A-treated junctions, the occurrence of polyneuronal innervation was extremely rare in control muscles (1/186 junctions, 0.53% of the fibers examined, in N = 4 human donors). Similarly, the incidence of nerve sprouting was exceptional among the NMJs examined, from patients that had an upper blepharoplasty and a similar age to the blepharospasm patients treated with BoNT/A. In only one out of 186 control NMJs examined (N = 4 human donors), two short ultraterminal nerve sprouts were detected, as illustrated in Figure 6F.

### 2.4. Statistical Analysis of Mono- and Poly-Innervated Neuromuscular Junctions

The marked differences between the BoNT/A-treated and control NMJs were clear and obvious, as shown in Figure 2, Figure 3, Figure 4, Figure 5 and Figure 6. To strengthen the validity of these findings, we performed a blind statistical analysis of the polyinnervation present in NMJs from control and BoNT/A-treated patients using the non-parametric Kruskal-Wallis test.

As shown in Figure 7A, the use of the non-parametric Kruskal-Wallis test permitted the two treatments with abobotulinumtoxinA (Dysport^®^) and with incobotulinumtoxinA (Xeomin^®^) to be compared with the control group, without assuming that values are normally distributed. Under these conditions, the percentage of polyinnervated NMJs in the two BoNT/A-treated patient groups was highly statistically different from that of the control group of the patient (*p* = 0.0029). Similarly, the use of the non-parametric Mann-Whitney U test allowed comparing the percentage of polyinnervated NMJs from all patients (N = 14) treated with BoNT/A-containing pharmaceutical forms with the control group of patients without assuming that values are normally distributed. As shown in Figure 7B, again, the difference between the two groups was highly significant, *p* = 0.003.

Single NMJs from BoNT/A treated patients were innervated by 1–6 motor axons, while the majority of control NMJs (99.47 ± 1.06%) were innervated by 1 motor axon (Figure 8). Mono-innervated junctions represented 25.16 ± 6.95% of all NMJs analyzed from BoNT/A-treated patients (Figure 8, green and yellow columns). Junctions innervated by 2 motor axons were the majority in BoNT/A-treated patients (36.03 ± 10.93%), while junctions innervated by 3 axons represented 26.31 ± 6.14%, and those innervated by 4 to 6 axons 12.50 ± 4.57%. It is worth noting that NMJs innervated by 3 or more axons were never observed in junctions from control patients (Figure 8).

### 2.5. Morphological Signs of Synapse Elimination in BoNT/A-Treated NMJs

Some morphological correlates of synapse elimination were detected in the *Orbicularis oculi* muscle from two out of fourteen blepharospasm patients (14.28%) responding poorly to BoNT/A treatment.

In the BoNT/A-treated muscles, the spatial intramuscular axonal distribution revealed segregated axons, which was characteristic of the presence of thin separated axonal branches that ended in distal swollen tip structures (Figure 9A–D). In addition, synaptic rearrangements were visualized at single NMJs, as depicted in Figure 9A,B and Figure 10. The presence of the so-called “retraction bulbs” at the tip of axons and nerve terminals was reminiscent of those described during the transition from multiple to single innervation at developing rodent neuromuscular junctions [51,52].

At human BoNT/A-treated NMJs, the retraction bulbs likely represent the common morphological sign of motor nerve terminals (Figure 9A,B) or intramuscular axons undergoing elimination (Figure 9D). It should be noted that retraction bulbs within NMJs had similar morphological characteristics to terminal bulbs of intramuscular axons.

A striking feature was the loss of nAChR staining in the muscle fiber membrane, specifically underneath the retraction bulbs (Figure 9A,B). At some endplates, the small and reduced nAChR patches were faintly stained (Figure 9A,B and Figure 10). Interestingly, nAChRs seemed to be removed earlier than the overlying retraction bulbs from motor nerve terminals.

## 3. Discussion

The use of confocal laser scanning microscopy and fluorescent immunostaining allowed us to study surgical-waste *Orbicularis oculi* muscle fragments from blepharospasm patients responding poorly to BoNT/A and from aged-matched control patients naïve of the toxin. The quantitative morphological study revealed that the innervation pattern of motor nerve terminals and the NMJs were quite different and more complex in muscles treated with repeated injections of BoNT/A than in control human *O. oculi* muscle fragments. These results are consistent with previous studies reporting a significant increase in the number of nerve fibers and NMJs in human *O. oculi* muscles from blepharospasm patients treated with repeated BoNT/A injections when compared to untreated control muscles [53]. Similar effects have been described following BoNT/A injection into rabbit eyelids, in which the histological density of NMJs was increased in the weeks following toxin treatment [54]. Such changes can be related to the motor nerve sprouting and the synaptic remodeling characterizing murine skeletal muscles treated with BoNT/A toxin [21,55,56,57,58,59,60]. Prior investigations assessing the morphology of single NMJs in patients with blepharospasm and recurrent BoNT/A injections revealed nerve sprouting of motor axons and nerve terminals in the removed *Orbicularis oculi* muscles [61]. Furthermore, in humans, the permanency of nerve terminal sprouts for three years has been recognized after repeated BoNT/A injections [62]. 

A conspicuous finding in the results obtained in the present study was that BoNT/A-treated muscles exhibited a stable polyneuronal and convergent innervation of the muscle fibers. Indeed, multiple axons innervating a single endplate or multiple endplates innervated by different axons on individual muscle fibers were prominent features in the morphological observations (Figure 2, Figure 3, Figure 4, Figure 5, Figure 6, Figure 7 and Figure 8). In marked contrast to control *Orbicularis oculi* muscles that were consistently innervated by a single axon (Figure 6, Figure 7 and Figure 8), as it has been reported previously in the *Orbicularis oculi* muscle [61], and in other human skeletal muscles [63,64,65].

In the present work, we found that polyneuronal innervation had an extremely rare occurrence (0.53%) in the NMJs of the control muscles studied, which is in good agreement with previous work done in human dissected *Orbicularis oculi* myofibers [61,66].

Developmental polyneuronal innervation has been described for the first time by J.F. Tello in 1907 [67] (a student and collaborator of Santiago Ramón y Cajal) in immature animals. Later, this process was rediscovered in newborn rat diaphragms in the 1970s [68] and in other skeletal muscles (reviewed in [69]). Since then, polyneuronal innervation has been demonstrated by visualizing motor axons and NMJs with histological techniques and/or by recording intracellular compound endplate potentials evoked by nerve stimulation [69,70]. In rodents, during the early stages of innervation, the axonal terminals from a number of motor neurons form numerous junctions on a small area (motor endplate) of every muscle fiber. Afterward, as maturation progresses, the amount of neuronal inputs at the endplate region markedly diminishes so that, at mature junctions, each muscle fiber is mono-innervated by terminals of a unique motor neuron. In rodents, polyneuronal innervation of muscle fibers is removed within the first three weeks of perinatal maturation (reviewed in [70,71,72]).

All cellular components of mammalian NMJs, including the presynaptic motor nerve terminals, the non-myelinating perisynaptic Schwann cells, and the postsynaptic skeletal muscle fibers, take part in the complex and multifactorial, and most probably polygenic processes involving synapse elimination during development [70,71,72].

The enhancement of activity by either nerve or muscle stimulation decreases polyneuronal innervation at the NMJ [73,74,75]. In contrast, the skeletal muscle paralysis produced by BoNT/A delays the removal of polyneuronal innervation, and paralyzed muscles by the toxin become accessible to innervation by a foreign nerve, comparable to what occurs in denervated muscles [75,76]. Similarly, the block of nerve conduction by tetrodotoxin was reported to lengthen synapse elimination, as well as the block of postsynaptic nAChRs with α-bungarotoxin or d-tubocurarine (through a postsynaptic action) also delayed the elimination of developmental polyneuronal innervation (reviewed in ref. [76]). Thus, during development, the structures composing the NMJ undergo rapid formation and elimination of unnecessary junctions (reviewed in [70,71,72,75,76,77,78].

Polyneuronal innervation can also occur in adult rodent skeletal muscle during the reinnervation of a muscle by its own nerve (following nerve crush) or during cross-innervation by a foreign nerve in an ectopic region after sectioning the original nerve [69,70,71,72]. In an adult rat model of ischemic stroke, an increased frequency of polyaxonal innervation has been reported at paretic NMJs. Interestingly, mechanical therapy prevented the stroke-induced rise in polyaxonal innervation [79].

To date, a few studies have been performed with repeated BoNT/A injections in vivo in experimental animals [80]. Similarly, there have been few reports on in vivo BoNT/A injections and ex vivo analyses of single NMJs. This is in part due to the abundant proliferation of sprouts in motor axons and nerve terminals, enlarging their field of innervation over the muscle fibers [58], thus making problematic the visualization of the presynaptic component of each single NMJ. When two or three BoNT/A injections were administered to mouse epitrochleoanconeus muscle at three to four month intervals, the recovery of quantal release to normal values was found to occur more slowly than after a single injection [55]. Furthermore, it was shown, using extracellular focal current recordings, that during the recovery from BoNT/A action, the area of the original NMJ released considerably more quanta than the newly formed terminals. This would not be surprising if polyneuronal convergent innervation of NMJs was present under those conditions. Even when no morphological evidence for polyneuronal innervation was reported at the NMJs, the fact that populations of endplate potentials were recorded upon nerve stimulation [55] gives support to the presence of polyinnervation following two or three BoNT/A injections in mouse NMJs.

The perisynaptic Schwann cells are involved in the removal of the superfluous junctions and are known to participate in synapse elimination but are not involved in selecting the motor axon that is finally maintained on each individual muscle fiber (reviewed in [70,71,72,77,78].

Only limited knowledge is available on polyneuronal innervation and its regression in developing human skeletal muscles (*quadriceps femoris* and *psoas* muscles [81,82,83].

Our results support the view that changes in the innervation of the *Orbicularis oculi* muscle may occur in mature human NMJs in response to repeated injections of BoNT/A. Signs of synapse elimination were observed in NMJs (Figure 9 and Figure 10) of a small proportion (14.28%) of muscles from the BoNT/A-treated patient responding poorly to the toxin. Our results raise an obvious question of why signs of synapse elimination were not more frequently observed in the BoNT/A-treated NMJ specimens. Whether this small proportion detected may reflect some heterogeneity in the patient’s blepharospasm pathology requires further exploration.

A striking feature in those mature junctions was the fact that postsynaptic nAChRs withdrawal occurred at a time when retraction bulbs in motor nerve terminals were still present (Figure 9). These results suggest that the loss of nAChRs may precede that of nerve terminals and point to a distinct signaling mechanism for the pre- and post-synaptic components of the NMJ. Furthermore, these results are in marked contrast to what is observed in permanently denervated mouse muscles that do not lose postsynaptic nAChR areas for long periods of time [84]. Similar effects have been reported in human denervated skeletal muscle in which nAChRs of the motor endplate persisted for more than six months after nerve injury [85].

Whether the maintenance of the polyneuronal innervation is due to the BoNT/A treatment or related to the inherent evolution of the blepharospasm-pathology of the patients remains to be determined. Whatever the origin is, it is worth noting that no difference was detected in the *Orbicularis oculi* muscle specimens from patients who received either abobotulinumtoxinA (Dysport^®^)- or incobotulinumtoxinA (Xeomin^®^)-treatment. In addition, it should be noted that the severity of the blepharospasm has been reported, in a 5-year longitudinal follow-up study, to increase consistently in patients aged 68–73 years [86]. The worsening is manifested in the patients by an increased occurrence of muscle spasms of long duration. Thus, aging seems to play a role in the progression of blepharospasm.

The pathophysiological origin of the reduction in clinical efficacy of BoNT/A remains unclear.

From a physiological point of view, one can anticipate that the functioning of a muscle fiber controlled by a single axon should be significantly different from that of a fiber innervated by several axons. In fact, such individual motor axons may asynchronously fire synaptic potentials (endplate potentials) and action potentials (due to distinct nerve conduction parameters, depending on their maturity) and evoke neurotransmitter release at the terminals, which in turn would trigger asynchronous muscle action potentials and ultimately asynchronous muscle contractions. Definitely, such type of asynchronous contractions would be physiologically unfavorable when compared to phasic contractions occurring under normal conditions.

Our study has several limitations. Primarily, these limitations are due to the fact that we have not included a group of patients with essential blepharospasm that have never been treated with BoNT/A. Due to ethical problems related to performing the surgical myectomy before knowing if BoNT/A is active in the patient, this group of patients is lacking in our study. However, it is worth noting that in a previous report on *Orbicularis oculi* muscle specimens from two patients with blepharospasm who were never injected with BoNT/A, it was shown that the observed nerve terminals and motor endplates were similar to normal control *Orbicularis oculi* muscles [61].

In addition, to disclose whether the changes here reported at the NMJ may be a direct effect of repeated BoNT/A injections or related to the lack of response to the treatment. It would be ideal to include a group of BoNT/A-treated patients who continue responding to the toxin. However, such type of study is not ethically feasible.

An additional limitation was the small number of control patients in our study, which is, in part, due because the ophthalmology department is not a referral center for aging or aesthetic eyelid problems and because it was difficult recruiting control patients matching the age of the BoNT/A-treated blepharospasm patients. Despite that only four control patients were recruited, the marked differences between the BoNT/A-treated and control NMJs (see Figure 7 and Figure 8) were well-defined and highly statistically significant using the non-parametric Kruskal-Wallis test and Mann-Whitney *U*-test.

To the best of our knowledge, the results here reported for the first time raise a number of questions on the origin and factors contributing to the synaptic plasticity changes observed, and in particular on synapse elimination. Synaptic plasticity at the NMJ depends in part on perisynaptic Schwann cells, which play an important role in developmental polyinnervation elimination, and in the sprouting process. The fact that BoNT/A treated junctions have a stable polyneuronal innervation strongly suggests that the Schwann cell is unable to eliminate polyneuronal innervation in NMJs.

Important questions remaining to be addressed include:(i)What are the cell signaling mediators allowing both polyneuronal innervation and the formation of multiple endplates in a single mature skeletal muscle fiber treated with BoNT/A?(ii)Does polyneuronal innervation affect the pattern of motor neuron activity, synaptic transmission, and skeletal muscle contraction in BoNT/A-treated muscles?(iii)How is synapse elimination altered in polyneuronal innervated BoNT/A treated muscles, and are the number and activity of the perisynaptic Schwann cell in situ modified in polyinnervated BoNT/A-treated NMJs?

These will be fruitful areas of work for the future. The answer to these questions will certainly expand our understanding of why, in a low percentage of cases, BoNT/A is no longer effective in treating blepharospasm in humans.

## 4. Conclusions

The present study provides novel insight into the presynaptic morphological characterization of axons and nerve terminals, as well as of postsynaptic nAChR of junctions from blepharospasm patients responding poorly to repeated injections of BoNT/A.

Our main conclusions are the following:(i)Changes in the innervation pattern of *Orbicularis oculi* muscle were present in mature single NMJ in response to treatment with repeated BoNT/A injections.(ii)A conspicuous finding of BoNT/A-treated *Orbicularis oculi* muscles was a stable polyneuronal convergent innervation of single muscle fibers, in marked contrast to control muscle fibers that were consistently monoinnervated.(iii)It remains to be determined if the polyneuronal innervation detected is due to the BoNT/A treatment or is related to the inherent patient’s blepharospasm pathology. Whatever the origin is, no difference was detected in the muscle specimens from patients receiving either abobotulinumtoxinA (Dysport^®^)- or incobotulinumtoxinA (Xeomin^®^)-treatment.(iv)The capacity to sprout new neuronal processes is still present and is not suppressed in patients with blepharospasm treated with repeated BoNT/A injections, for which the toxin is less effective.(v)Signs of synaptic elimination were observed in BoNT/A-treated muscles as evidenced by the presence of retraction bulbs, which are a recurrent morphological sign of motor nerve terminals undergoing elimination within the NMJ domain and intramuscular axons, but most probably the pruning of axons and nerve terminals cannot cope, suggesting alterations in perisynaptic Schwann cell activity and signaling.

## 5. Materials and Methods

### 5.1. Patients with Blepharospasm

A small number of patients suffering from essential blepharospasm and treated at the ophthalmology department of the Tenon Hospital responded poorly to BoNT/A treatment. It should be noted that none of the patients in question exhibited any signs of “resistance” to BoNT/A or “non-responsiveness” to BoNT/A injections. In specific cases, their muscles were abnormally stimulated by very disabling dystonia, requiring more BoNT/A units with a shortened reinjection time. These patients were clinically examined, and blood samples were obtained for testing the absence of BoNT/A neutralizing antibodies. Additionally, the effectiveness of BoNT/A was evaluated by injecting the BoNT/A into the upper part of the hemi-frontalis muscle. Patients were examined ten days later by comparing the injected with the contralateral non-injected hemi-frontalis muscle. Once these two tests were carried out (absence of neutralizing antibodies and effectiveness of the toxin on a muscle not affected by the disease), a BoNT/A injection session was conducted. The BoNT/A concentration was doubled, and an increased number of injection points were used. The standardized “Jankovic Rating Scale” [87] was used to assess the BoNT/A treatment response in patients with blepharospasm. The two blepharospasm parameters, severity, and frequency (both scored 0–4), were evaluated. Patients to whom myectomy surgery was proposed had a score of 4 in blepharospasm severity (severe, incapacitating spasm of eyelids) and a score of 4 in blepharospasm frequency (functionally blind due to persistent eyelid closure for more than 50% of the waking time). In cases where the BoNT/A concentration needed to be doubled and/or injection points needed to be multiplied, associated with a shortening of intervals between two re-injection times, the partial upper *Orbicularis oculi* myectomy was considered for improving the treatment. In those cases, the myectomy surgery was explained to patients, including a comprehensive overview of the potential benefits and risks. This outdoor surgery was performed under local anesthesia. Scheduled BoNT/A re-injections after surgery were also explained to the patient. Under any circumstances, it was suggested that the surgery could cure the disease. Once the patient had provided informed consent for the surgery, they were informed of the use of surgical waste muscle fragments for immuno-histology research. The present study included patients who had approved the proposed therapeutic strategy. Those patients who refused surgery were not included in the study. Furthermore, patients who underwent a treatment change in the BoNT/A pharmaceutical form prior to the surgical myectomy were not included in the study. After their consent, the surgery was performed, and muscle waste samples were prepared, as detailed here below.

### 5.2. Orbicularis oculi Surgical-Waste Muscle Fragments

*Orbicularis oculi* muscle fragments were obtained from fourteen patients undergoing surgical myectomy for essential blepharospasm at the Tenon Hospital (Paris, France). In addition, four patients undergoing upper blepharoplasty (eyelid lift) to cure unaesthetic ptosis and who had never been exposed to BoNT/A and had no focal dystonia (blepharospasm) or antecedents of neuromuscular diseases were considered as controls.

The blepharospasm patients had been previously injected, for varying times, with two commercially available BoNT/A pharmaceutical products, either abobotulinumtoxinA (Dysport^®^, Ipsen Ltd., Berkshire, UK) or incobotulinumtoxinA (Xeomin^®^, Merz, Pharmaceuticals GmbH, Frankfurt, Germany) and the last toxin injection ranged from four days to three months before myectomy. Each injection consisted of 0.1 mL of either incobotulinumtoxinA (100 Xeomin units/mL) or abobotulinumtoxinA (250 Speywood units/mL) administered in internal and external upper pretarsal and temporal upper preseptal *Orbicularis oculi* muscle, for the area involved by eyelid surgery. The surgical removal of upper *O. oculi* muscle fragments was always performed by B.G., one of us, and is described in some detail in Appendix A.

The waste muscle samples obtained by the surgery were immediately pinned flat on silicon-embedded Petri dishes at the resting length and washed repeatedly first, with a physiological mammalian oxygenated Krebs-Ringer solution containing (in mM) 135 NaCl, 5 KCl, 2 CaCl_2_, 2 MgCl_2_, 10 glucose, and 10 HEPES buffer solution (adjusted to pH 7.4). Thereafter, muscle fragments were washed with a solution containing Phosphate Buffer Saline (PBS, Thermo Fisher 14190-144, Illkirch, France) to eliminate the remaining blood adhering to samples. Fixation was done with 4% paraformaldehyde (Electron Microscopy Sciences, Hatfield, PA, USA) in PBS for 30 min at room temperature (20–22 °C) and transferred to the laboratory. After two subsequent PBS washes, autofluorescence was quenched with 50 mM NH_4_Cl (Euromedex, Souffelweyersheim, France) for 30 min, and muscle fragment samples were rinsed in PBS before dissection under a binocular microscope to carefully remove connective and fat tissues, and therefore bring-down possible background staining. Small bundles of fibers were teased out from the muscle fragments for whole-mount preparation.

### 5.3. Nerve Terminal Immuno-Labelling and Muscle Staining

Immuno-fluorescence labeling or staining was achieved on whole-mount micro-dissected muscle fragment preparations permeabilized by incubation for 2 h with 2% Triton X-100 (Sigma-Aldrich, Saint-Quentin-Fallavier, France) in PBS and blocked by incubation for 2 h at room temperature with 2% bovine serum albumin (Sigma-Aldrich) (permeabilization/saturation buffer). Muscle preparations were then processed with anti-neurofilament 160/200 kDa antibodies (mouse monoclonal, Sigma-Aldrich, 1:100 dilution) and anti-β-III tubulin (Rabbit monoclonal, Tuj1, BioLegend, San Diego, CA, USA; MRB 435P, 1:200 dilution) incubated overnight at 4 °C in permeabilization/saturation buffer). After three PBS washes, samples were incubated overnight at 4 °C with Alexa Fluor 555 labeled goat anti-mouse IgG (A-21422, 1:500 dilution, Thermo Fisher, Illkirch, France), Alexa Fluor 647 labeled goat anti-rabbit IgG (A-21244, 1:500 dilution, Thermo Fisher), Alexa Fluor 488 labeled α-bungarotoxin (B-13422, 20 µg/mL, Thermo Fisher), and 4′,6-diamidino-2-phenylindole (Sigma-Aldrich, 5 µg/mL). After three PBS washes, muscle samples were flat-mounted in Mowiol^®^ 4-88 mounting medium (Sigma-Sigma-Aldrich) between the slide and coverslip and kept at 4 °C before imaging.

### 5.4. Image Acquisition and Processing

Images were acquired with multiphoton scanning confocal microscopes Zeiss LSM 510 META (Carl Zeiss, Jena, Germany) or Leica TCS SP8 (Leica Mikrosysteme Vertrieb GmbH, Wetzlar, Germany) and controlled through the manufacturer-supplied software and workstations. The confocal images presented herein comprise single projected images obtained by superimposing collected z-stack sets. Z-stack images (1024 × 1024 pixels) were obtained from a series of optical sections (0.5 μm) with a Plan-Apochromat 10 x/0.4 numerical aperture (NA), or 63 x/1.4 NA oil objectives, and projections were made and analyzed using the Java image processing software ImageJ version 1.54 (https://imagej.nih.gov/ij/; accessed on 23 January 2023). Polyinnervation was assessed by counting the number of independent axonal inputs contacting a single NMJ endplate. Individual NMJs were then classified as mono- or poly-innervated.

### 5.5. Statistics and Data Processing

All data was managed anonymously. Numbers and colors replaced patient’s names and BoNT/A containing pharmaceutical form names, respectively. Unless otherwise stated, data are presented as the mean ± SD. Comparison between data was completed using the non-parametric Kruskal-Wallis test that allowed the two treatments, either with abobotulinumtoxinA (Dysport^®^) or incobotulinumtoxinA (Xeomin^®^), to be compared with the control group without assuming that values are normally distributed. In addition, the Mann-Whitney U-test was used to compare BoNT/A-treated patients with the control group. Differences were considered statistically significant at *p* < 0.05. N represented the number of donor patients; n was the number of junctions examined. Statistical analyses were performed using GraphPad Prism 9.0 software (GraphPad Software Inc., San Diego, CA, USA).

## Figures and Tables

**Figure 1 toxins-16-00506-f001:**
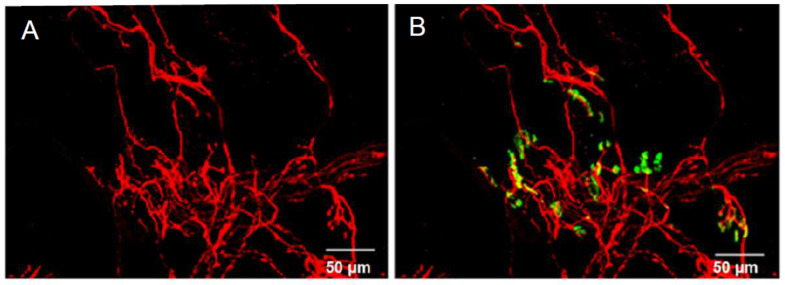
Confocal micrographs of the innervation pattern and the endplates in a surgical-waste *Orbicularis oculi* mount-muscle fragment from a patient treated with BoNT/A for 1 year. In (**A**) are shown intramuscular axons and nerve terminals immunostained for neurofilaments (in red), and in (**B**) are shown both neurofilaments (in red) and nAChRs stained with α-bungarotoxin conjugated with fluorescein (in green).

**Figure 2 toxins-16-00506-f002:**
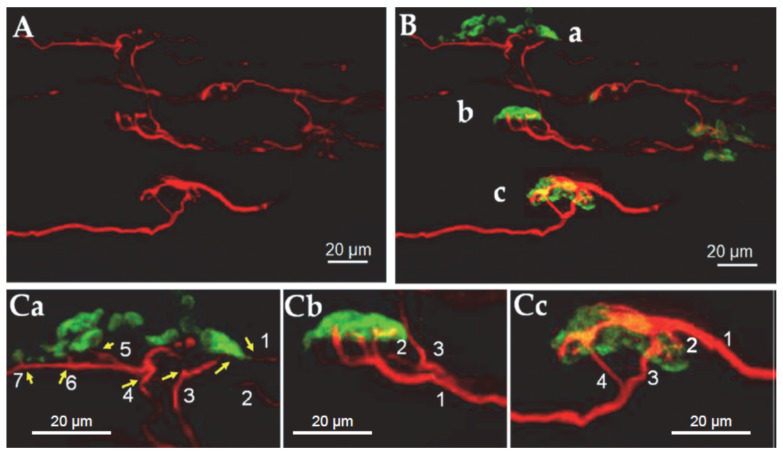
The pattern of innervation and endplates imaged at high magnification with a confocal microscope from a small part of the *Orbicularis oculi*-mount surgical waste muscle fragment shown in Figure 1. In (**A**), axons and nerve terminal immunostained for neurofilaments (in red). Shown in (**B**) are immunostained axons and terminals innervating three different endplates labeled with fluorescent α-bungarotoxin (a, b, c, in green). Shown in (**Ca**,**Cb**,**Cc**) are enlarged (2.0 times) images of (**B**), note the polyinnervated endplates. The numbers refer to the axonal inputs (yellow arrows) to the same endplate. Note the different axonal calibers converging to single NMJs.

**Figure 3 toxins-16-00506-f003:**
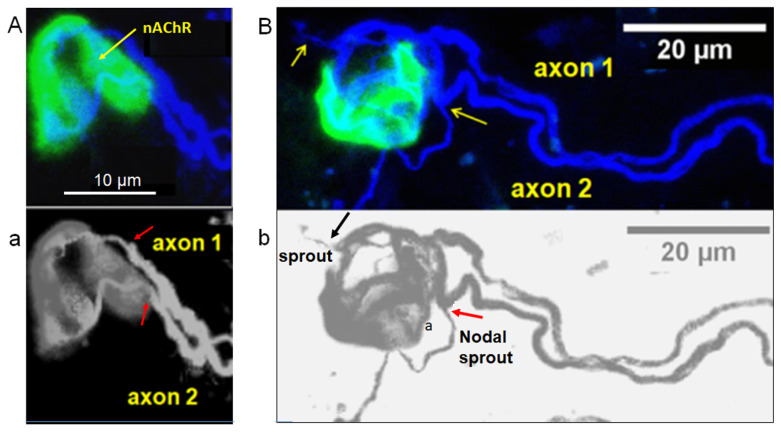
Examples of stable NMJs innervated by two motor axons in BoNT/A-treated *Orbicularis oculi* muscles. In (**A**,**a**), two myelinated axons with distinct caliber (labeled 1 and 2) converge to a unique endplate exhibiting relatively homogeneous staining with fluorescent α-bungarotoxin to label the nAChRs. Note, in (**a**) the heminode region (end of the myelin sheath, red arrows) of both axon 1 and axon 2. Same calibration in (**A**,**a**). (**B**,**b**) displays another complex NMJ innervated by two distinct axons. Note, in (**B**,**b**), several thin nerve terminals budding from the two axons, an ultraterminal nerve sprout (axon 1, yellow and black arrows, respectively), and a nodal sprout (axon 2, yellow and red arrows, respectively).

**Figure 4 toxins-16-00506-f004:**
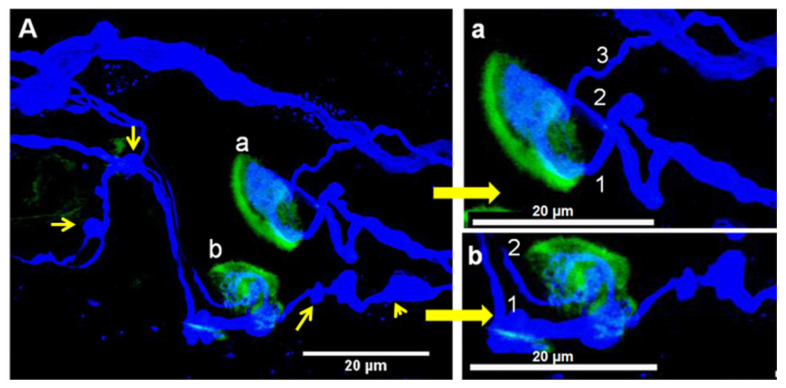
(**A**) Confocal micrograph of a representative example of convergent innervation in an *Orbicularis oculi* muscle fragment immunostained for neurofilaments and tubulin (blue) and stained with fluorescent α-bungarotoxin conjugated with fluorescein (yellow). The patient was treated for 1 year with BoNT/A, and the last toxin injection was performed 6 weeks before the partial myectomy. The two NMJ in (**A**), labeled (**a**,**b**), are shown at higher magnification in panels (**a**,**b**). The numbers 1–3 in (**a**) and 1, 2 in (**b**) refer to the axonal inputs to the respective endplates. Note in (**A**) the presence of oval structures connected to axons (yellow thin arrows) that may represent signs of synaptic elimination and enlarged axonal areas (yellow arrowhead).

**Figure 5 toxins-16-00506-f005:**
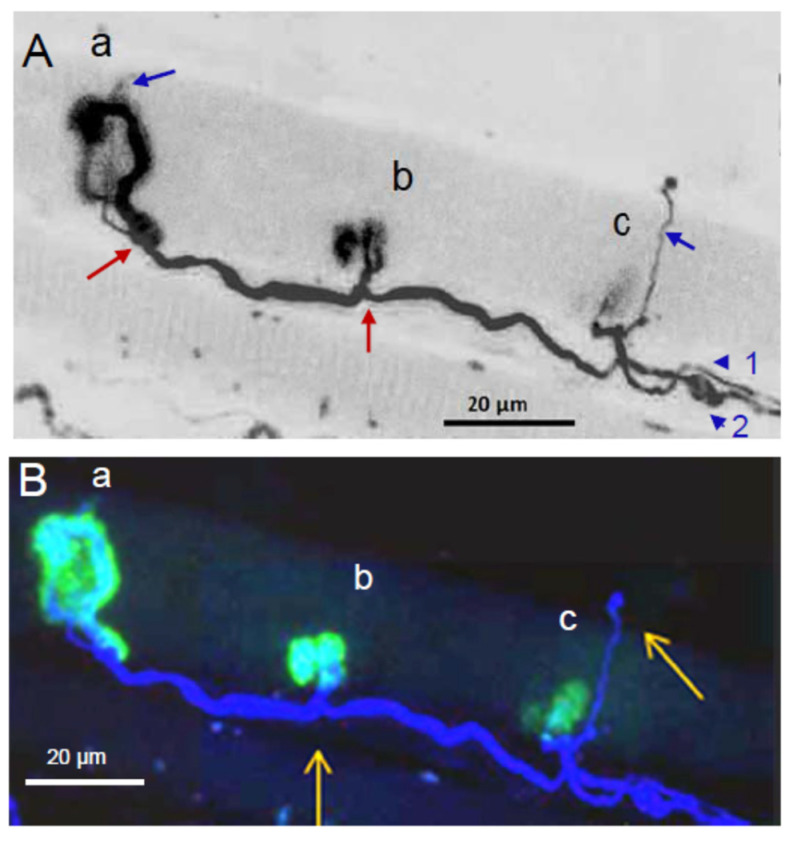
Confocal micrographs of the innervation pattern in a surgical-waste *Orbicularis oculi* mount-muscle fragment, collected 6 weeks after the latest toxin injection, from a patient treated with BoNT/A for 4 years. In (**A**,**a**–**c**) are shown two axons (blue arrowheads 1 and 2) innervating the same muscle fiber at distinct sites. In (**a**), note the thin nodal sprout emerging from the enlarged pre-terminal axon (red arrow) and the small terminal sprout (blue arrow). In (**b**) is also shown a nodal sprout (red arrow) evolving from the main nerve trunk 2. Note in (**c**), another axon innervating the same muscle fiber and the collateral sprout (blue arrow), budding from a nerve terminal branch. Note at the tip of the collateral sprout, there is an oval structure resembling a retraction bulb. In (**B**,**a**–**c**), the same image as in (**A**), showing the fluorescent staining with α-bungarotoxin and the different extension and intensity of the labeled nAChRs (**a**–**c**). The two yellow arrows show nodal and collateral sprouts on the same muscle fiber.

**Figure 6 toxins-16-00506-f006:**
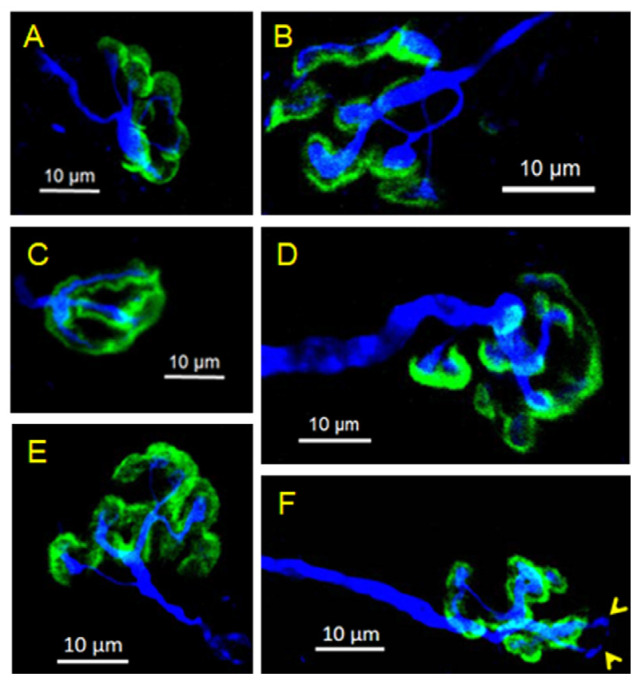
(**A**–**F**) Confocal micrographs of the innervation pattern of *Orbicularis oculi* mount-muscle fragment collected from control patients that were submitted to upper blepharoplasty. These patients had neither a history of neuromuscular diseases nor blepharospasm and had never been injected with BoNT/A. Note the variability in NMJ profiles and the distinct extension of nAChR clusters labeled with fluorescent α-bungarotoxin (in green). In (**F**), note the occurrence of two ultraterminal short nerve sprouts (yellow arrowheads).

**Figure 7 toxins-16-00506-f007:**
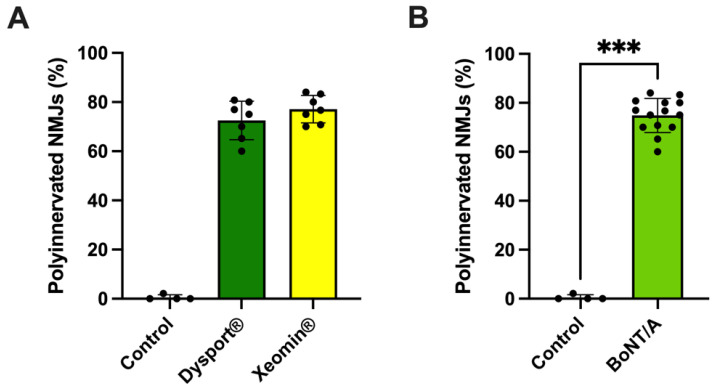
Percentage of polyinnervated NMJs from control patients and from patients treated with BoNT/A. (**A**) Percentage of polyinnervated NMJs from control patients (N = 4 donors; n = 186 NMJs examined) and from patients treated either with abobotulinumtoxinA (Dysport^®^, green column; N = 7, n = 160 NMJs) or with incobotulinumtoxinA (Xeomin^®^, yellow column; N = 7; n = 195 NMJs). Each black circle in columns represents mean data sampled from 1 patient. Data are presented as the mean ± SD. The difference between groups was statistically significant (*p* = 0.0029; Kruskal-Wallis test). (**B**) Percentage of polyinnervated NMJs from control patients (N = 4 donors; n = 186 NMJs examined) and from patients treated with abobotulinumtoxinA (Dysport^®^) and with incobotulinumtoxinA (Xeomin^®^) (Yellow Green column; N = 14, n = 355 NMJs). Data are presented as the mean ± SD. The difference between the control and BoNT/A-treated polyinnervated NMJ group was statistically significant (***) *p* = 0.0003); Mann-Whitney U-test.

**Figure 8 toxins-16-00506-f008:**
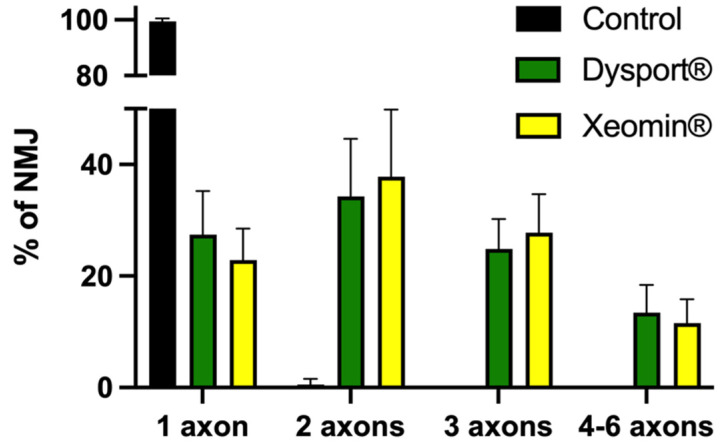
Percentage of NMJs innervated by 1 to 6 motor axons in control patients (N = 4 donors, black column), and in patients treated with abobotulinumtoxinA (Dysport^®^) (N = 7; green columns), and with incobotulinumtoxinA (Xeomin^®^) (N = 7; yellow columns. Data is expressed as the mean ± SD. Note the significant difference between control and BoNT/A-treated NMJs innervated by 1 and 2 axons and the absence of innervation by 3 and 4–6 axons in control NMJs. No significant statistical difference was detected between the two pharmaceutical brands studied containing BoNT/A.

**Figure 9 toxins-16-00506-f009:**
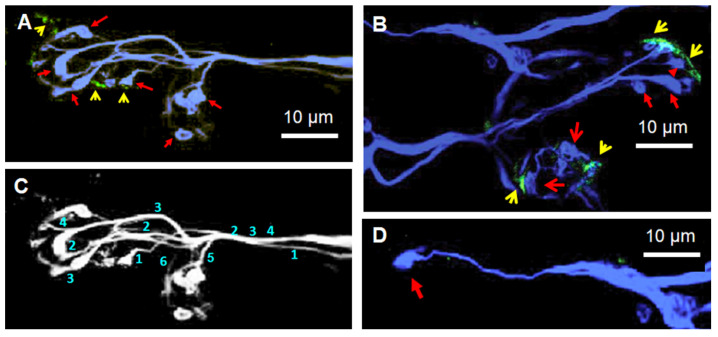
(**A**–**D**) Signs of synaptic elimination in BoNT/A-treated *O. oculi* NMJs. (**A**,**C**) Presence of retraction bulbs (red arrows) with thin axonal terminals in close proximity to small clusters of nAChRs (yellow arrowheads). Note that from the six retraction bulbs, only three of them show a small nAChR cluster. (**C**) In the same image as in (**A**), the numbers indicate the axonal count in the synaptic region. (**B**) Two polyinnervated NMJs from another BoNT/A-treated patient showed nAChR clusters (yellow arrowheads) and some retraction bulbs, some of which are devoid of nAChR clusters (red arrows). (**D**) Unique retraction bulb (red arrow) attached to an atrophied thin axonal extension emerging from a polyaxonal intramuscular nerve trunk.

**Figure 10 toxins-16-00506-f010:**
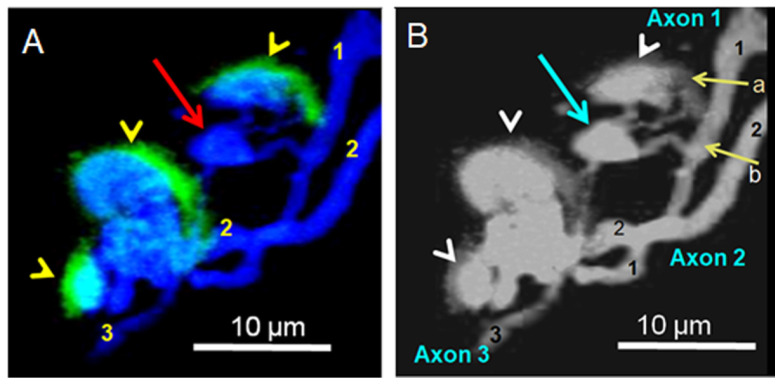
(**A**,**B**) Single NMJ innervated by three distinct axons (numbered 1–3) showing a retraction bulb (red and turquoise arrows in (**A**,**B**), respectively) located in between innervated nAChR clusters stained with fluorescent α-bungarotoxin (yellow arrowheads). Note in (**B**) that the retraction bulb is connected to at least two nodal axonal branches emerging from the axon 1 (indicated by the yellow arrows labeled a and b, respectively).

## Data Availability

The original contributions presented in the study are included in the article, further inquiries can be directed to the corresponding author.

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
