# Peer review of "Stable Convergent Polyneuronal Innervation and Altered Synapse Elimination in Orbicularis oculi Muscles from Patients with Blepharospasm Responding Poorly to Recurrent Botulinum Type-A Neurotoxin Injections"

_toxins, 2024, doi:10.3390/toxins16120506_

Round 1
Reviewer 1 Report
Comments and Suggestions for Authors
This study investigates the effects of repeated Botulinum neurotoxin type-A (BoNT/A) injections on the neuromuscular junctions (NMJs) of patients with blepharospasm who no longer responded to treatment. Muscle samples from 14 patients treated with Dysport® or Xeomin® were compared to those from of 4 untreated individuals. Confocal microscopy revealed differences in the innervation patterns and NMJ structures of BoNT/A-treated muscles. Unlike the simple, mono-innervated NMJs in control samples, treated muscles exhibited complex polyneuronal innervation, with multiple axons converging on single muscle fibers. Additionally, the authors observed signs of impaired synapse elimination, such as retraction bulbs in axons and nerve terminals and a reduced extension of postsynaptic nAChRs.
The study topic is interesting, the paper is well-written and methodologically sound, and the figures are clear. However, there are some major critical issues that the authors should address before the paper could be considered for publication.
The first major point of criticism concerns the absence of quantitative measures and corresponding statistical analysis, even though this is briefly mentioned in the Methods section on page 13. As a first step, the clinical demographic data of the two groups (treated and untreated) should be compared using appropriate statistical methods to ensure the groups are comparable. Moreover, the authors have restricted their analysis to a qualitative description of the observed patterns in biopsy samples. To strengthen the findings, the introduction of quantitative measures would be valuable—for instance, assessing parameters such as the number of neuromuscular junctions per mm or nerve density. These metrics could then be compared between groups using statistical methods, even simple ones such as Fisher's exact test to compare frequencies.
A further criticism concerns the small sample size of the control group, which consists of only 4 individuals. The authors should either increase the sample size to strengthen the reliability of the results or, if this is not possible, explicitly acknowledge this limitation in the discussion section. Moreover, the number of healthy controls should be clearly stated in both the abstract and the results. Currently, the number (4) is only first mentioned on page 11, which may lead to confusion.
A key point that is only briefly discussed is the pathophysiological significance of the alterations observed in the patients. The changes at NMJ level may be a direct effect of repeated BoNT/A injections or related to the lack of response to the treatment. Ideally, including a group of treated patients who continue to respond to BoNT/A would enable the authors to determine whether the NMJ alterations are specifically associated with non-response. Again, it has been shown that botulinum toxin is active at multiple levels, namely sensory nerve terminals, and muscle spindles. Studies in animals have also shown that the toxin can be retrogradely transported and even transcytosed to neurons in the central nervous system (CNS). Further human studies have suggested that BT could alter the functional organisation of the CNS indirectly (10.1111/j.1468-1331.2010.03056.x; 10.1016/j.clinph.2011.05.032). These mechanisms should be mentioned as possible explanation of the results of the present study. Another alternative hypothesis is that the observed changes are part of the underlying pathological features of blepharospasm itself, as the authors briefly mentioned in the discussion. To better address this issue, it would be beneficial to include a group of patients with blepharospasm who have never been treated with BoNT/A, similar to the approach taken by Olson in their study (10.1080/02713683.2018.1543707). Although these comparisons may not be easily feasible for ethical and practical reasons, the authors should still discuss them in the manuscript.
It is unclear how the authors defined and identified non-responders among the study participants. Specifically, it would be helpful to know whether a standardized clinical blepharospasm rating scale was utilized to assess treatment response systematically. Furthermore, it is essential to address whether any patients underwent a treatment change in their scheme or toxin type prior to myectomy.
The authors may want to avoid using the term ‘extrapyramidal disorders’ on page 2, as it is considered outdated in contemporary neurological terminology (see doi: 10.1212/CPJ.0000000000200308)
There are some relevant references on the topic missing: see 10.1016/j.clinph.2024.02.023 concerning the physiology of blinking and 10.1212/wnl.41.11.1800 on botulinum-induced alteration of nerve-muscle interactions.
Other minor comments:
Page 8, line 276: please avoid the expression ‘quite different’
Page 8, line 278: please delete the comma after ‘reporting’.
Reviewer 2 Report
Comments and Suggestions for Authors
Toxins 3251629
The Title of this manuscript is too long and complicated. I recommend a much shorter version. For example:
Effects on orbicularis oculi muscle in patients responding poorly to botulinum toxin
Introduction
Citation 5 is too old and should be deleted. The first 4 citations deal with the subject adequately
Citations 6 & 7, relating to foodborne botulism, are not relevant to the manuscript and should be deleted
Line 44 The term “cosmetics” is incorrect and should be replaced with “aesthetic use”
Citations 20-22 are not required. Earlier citations used by the authors adequately describe the mode of action of BoNT/A.
Line 52 The term “remarkable” is not required
Lines 56-57 The part “….as reported…..” should be removed.
Citations 25-27 are not required. Earlier citations used by the authors adequately describe the mode of action of BoNT/A.
Lines 57-67 are not required and should be deleted along with the cited references. The authors article is not intended to be a review of the mode of action of botulinum toxin.
Lines 80-81 The part in () should have a citation
Line 81 The time between injections should have a citation
Citations 43-46 I recommend only 2 citations are needed here
Line 100 The authors should also include other published citations of their work on the same subject:
Girard, B., Couesnon, A., & Molgó, J. (2018). Synaptic remodeling of human orbicularis oculi muscles from patients treated with botulinum toxin type A. Toxicon, 156, S39-S40. https://doi.org/10.1016/j.toxicon.2018.11.096
Girard, B., Couesnon, A., Popoff, M.-R., & Molgó, J. (2018). Motor nerve sprouting and neuromuscular junction remodeling of orbicularis oculi muscles from patients with blepharospasm treated with botulinum neurotoxin type A. Toxicon, 156, S40. https://doi.org/https://doi.org/10.1016/j.toxicon.2018.11.097
Girard, B., Couesnon, A., Girard, E., & Molgó, J. (2022). Morphological Evidence for the Presence of Nerve Sprouting and Stable Poly-Innervation in Human Orbicularis oculi Muscles Treated With Repeated Injections of Botulinum Neurotoxin Type A. Toxicon, 214, S21. https://doi.org/https://doi.org/10.1016/j.toxicon.2021.11.053
Materials and Methods
Figure A2 and lines 530 & 532 The term “meticulous” should be deleted
The Informed Consent statement details that informed consent was obtained from all patients. But the Institutional Review Board Statement details that this was not applicable and ethical review and approval were waived. These are completely contradictory and should be explained by the authors.
Line 453 What does “mammalian” mean here?
Line 462 ”bring-down” should be written as “reduce”
Results
Lines 103-106 are not required, they are not results and are repetition
Lines 120-121 are not required – they are repetition
Lines 121-124 are not required – they are not results
Figure 1 There is no value to include this as higher magnification and detailed interpretation are given in Figure 2. I recommend Figure 1 be deleted.
Figures 2C and 3Aa need a scale bar adding to each picture
Lines 155, 172 et seq What does “calibers” mean?
Discussion
Lines 302-307 Citation needed
Lines 376-378 The authors should re-write this part since they have said that the severity deteriorated, which means an improvement in blepharospasm! They mean to say the severity increased……
Lines 381-384 This is entirely speculation. I recommend removal. For example, there is no evidence for a change in SNAP-25 cleavage by toxin with time. Any reduction in clinical efficacy is likely to be due to a decrease in available toxin targets……
The authors have not considered the normal, natural innervation of facial muscles, which is very different to other muscles in the body
Goodmurphy, C. W. (1996). A comparative morphological study of two human facial muscles: the orbicularis oculi and the corrugator supercilii [Masters, University of British Columbia]. Vancouver, Canada.
Goodmurphy, C. W., & Ovalle, W. K. (1999). Morphological study of two human facial muscles: orbicularis oculi and corrugator supercilii. Clin Anat, 12(1), 1-11. https://doi.org/10.1002/(SICI)1098-2353(1999)12:1<1::AID-CA1>3.0.CO;2-J
Comments on the Quality of English Language
Can be improved in places
Author Response
Please see attchment

Reviewer 3 Report
Comments and Suggestions for Authors
Abstract
Line 12: The brands Dysport® and Xeomin® are mentioned. You can use abo and incobotulinumtoxinA instead of brand names.
Line 14: "The morphological study was performed blinded to the BoNT/A sample treatment." Clarify whether the blinding was regarding the type of BoNT/A used or between treated and control samples.
Lines 16-19: The abstract mentions "marked differences" without specifying what they are. Briefly highlighting the key findings (e.g., polyneuronal innervation, convergent innervation) would strengthen the abstract.
Lines 20-22: The term "stable convergent innervation" may not be immediately clear to all readers. Consider rephrasing or providing a brief explanation.
Provide quantitative data in the abstract to support the statements (e.g., percentages of polyneuronal innervation observed).
Introduction
Lines 42-46: The therapeutic applications of BoNT/A are appropriately cited, but consider adding more recent references to ensure up-to-date information (e.g. https://doi.org/10.3390/toxins16040184, https://doi.org/10.3390/toxins16070309)
Lines 51-69: The detailed mechanism of BoNT/A is described. While informative, this section might be overly detailed for the introduction. Consider summarizing to focus on aspects directly relevant to the study.
Line 52: Consider summarizing the mechanism of BoNT/A to focus on its relevance to synapse elimination and neuromuscular junction alterations.
Methods
Lines 438-442: The sample collection is described, but more details are needed regarding inclusion/exclusion criteria.
Provide more information about the control group.
The authors mention that data were managed anonymously and that analyses were blinded. However, more details on the blinding process would strengthen this.
Elaborate on how blinding was maintained during sample analysis to ensure objectivity.
Lines 493-497: Statistical methods are briefly mentioned, but normality test are not detailed. Justify its use, and mention any software used for analysis.
Results
Line 105: "The mean number of toxin injections delivered to patients before palpebral surgery was 17.4 (range 4.5-40 per treated subject)." Clarify if "4.5" injections is correct, as partial injections are unusual.
Line 117: Specify the units clearly (e.g., units per injection).
Quantitative data on the frequency of polyneuronal innervation are lacking. Specifically, the manuscript does not mention: The total number of NMJs analyzed in BoNT/A-treated muscles. The number or percentage of NMJs that exhibited polyneuronal innervation. Statistical comparisons between the treated and control groups regarding the frequency of polyneuronal innervation.
Lines 215-216: "In only one out of 186 control NMJs examined (N = 4 human donors), two short ultraterminal nerve sprouts were detected..."
Line 216: The frequency of polyneuronal innervation in controls is given as 0.00537%. Provide raw numbers (e.g., 1 out of 186 fibers) for clarity.
The results focus on patients who did not respond to BoNT/A, which is the study's target population.
There is a lack of statistical analysis in comparing treated and control groups.
Discussion
Lines 312-326: The discussion on developmental polyneuronal innervation provides context but could be more focused on how it relates to the findings.
Limitations: Explicitly discuss the study's limitations
Explore potential mechanisms for altered synapse elimination in BoNT/A-treated muscles. Consider discussing the role of perisynaptic Schwann cells in more detail.
Discuss how these findings might influence clinical practice or the development of new treatments.
Conclusions
Emphasize the potential impact of the findings on understanding BoNT/A treatment failure and blepharospasm pathology.
Comments on the Quality of English LanguageLine 43: "Currently, a number of movement disorders are treated with BoNT/A (reviewed in ref. 14-19]." Incorrect punctuation with the reference (closing bracket should be before the period)
Line 117: "The mean number of BoNT/A units used per injection was 350.1 ± 64.5 for Dysport® (n =7) and 75.7 ± 9.96 (n =7) for Xeomin®." Inconsistent spacing and missing units.
Line 216: "The occurrence of polyneuronal innervation was extremely rare in controls muscles..." "Controls muscles" should be "control muscles."
Line 367-369: "Our results suggest that the loss of nAChRs may precede the one of nerve terminals, and point to a distinct signaling mechanism for the pre- and post-synaptic components of the NMJ." The phrase "the one of nerve terminals" is unclear.
Line 297-299: "In developing rodent skeletal muscles, during the early stages of innervation, the axonal terminals from a number of motor neurons form numerous junctions on a small area (motor endplate) of every muscle fiber." The sentence is long and can be split for clarity.
732
Round 2
Reviewer 1 Report
Comments and Suggestions for Authors
The authors have addressed my major comments. I believe the overall quality of the paper has improved, and it is now suitable for publication.
Author Response
We thank the reviewer for the valuable comments and suggestions
Reviewer 2 Report
Comments and Suggestions for Authors
The authors have rejected over half of the recommendations I made for improvements to the manuscript.
I refer this manuscript to the Editors now.
Comments on the Quality of English LanguageMinor improvements are required
Author Response

(The authors gave the same response as above.)

Reviewer 3 Report
Comments and Suggestions for Authors
Thank you for the authors responses and the revisions made to the manuscript. The efforts are appreciated, however, certain requests remain unfulfilled or inadequately explained.
Significant additional issues were outlined in the revised version.
Regarding the statistical analysis, the use of Fisher's exact test is not appropriate in this context. Fisher's exact test is suitable for small sample sizes and assumes that each observation is independent. In this study, multiple neuromuscular junctions (NMJs) from the same patient are analyzed, which means the observations are not independent because NMJs from the same patient are likely similar due to shared factors. Using Fisher's exact test in this situation can lead to incorrect conclusions. It is strongly recommended to reanalyze the data using appropriate statistical methods.
The conclusions may be overstated. Given the limitations in the control groups and the statistical analysis issues, the conclusion that BoNT/A treatment leads to stable polyneuronal innervation and altered synapse elimination may not be fully supported. Caution should be exercised in attributing effects solely to BoNT/A treatment without sufficient evidence.
Lastly, even if the clinical implications are not immediately clear, it is important to elaborate on how the findings might influence clinical practice or contribute to the development of new treatments for blepharospasm.
Inconsistency in used terminology and abbreviations
Author Response
Please see the second pdf file
